# A Novel Sensor Foil to Measure Ski Deflections: Development and Validation of a Curvature Model

**DOI:** 10.3390/s21144848

**Published:** 2021-07-16

**Authors:** Christoph Thorwartl, Josef Kröll, Andreas Tschepp, Philipp Schäffner, Helmut Holzer, Thomas Stöggl

**Affiliations:** 1Department of Sport and Exercise Science, University of Salzburg, Schlossallee 49, 5400 Hallein/Rif, Austria; josef.kroell@sbg.ac.at (J.K.); thomas.stoeggl@sbg.ac.at (T.S.); 2Joanneum Research Forschungsgesellschaft mbH, Franz-Pichler-Straße 30, 8160 Weiz, Austria; Andreas.Tschepp@joanneum.at (A.T.); Philipp.Schaeffner@joanneum.at (P.S.); 3Atomic Austria GmbH, Atomic Strasse 1, 5541 Altenmarkt, Austria; Helmut.Holzer@atomic.com; 4Athlete Performance Center, Red Bull Sports, Brunnbachweg 71, 5303 Thalgau, Austria

**Keywords:** bending sensors, flexion, PyzoFlex, ski bending, ski deflection

## Abstract

The ski deflection with the associated temporal and segmental curvature variation can be considered as a performance-relevant factor in alpine skiing. Although some work on recording ski deflection is available, the segmental curvature among the ski and temporal aspects have not yet been made an object of observation. Therefore, the goal of this study was to develop a novel ski demonstrator and to conceptualize and validate an empirical curvature model. Twenty-four PyzoFlex^®^ technology-based sensor foils were attached to the upper surface of an alpine ski. A self-developed instrument simultaneously measuring sixteen sensors was used as a data acquisition device. After calibration with a standardized bending test, using an empirical curvature model, the sensors were applied to analyze the segmental curvature characteristic (m^−1^) of the ski in a quasi-static bending situation at five different load levels between 100 N and 230 N. The derived curvature data were compared with values obtained from a high-precision laser measurement system. For the reliability assessment, successive pairs of trials were evaluated at different load levels by calculating the change in mean (CIM), the coefficient of variation (CV) and the intraclass correlation coefficient (ICC 3.1) with a 95% confidence interval. A high reliability of CIM −1.41–0.50%, max CV 1.45%, and ICC 3.1 > 0.961 was found for the different load levels. Additionally, the criterion validity based on the Pearson correlation coefficient was R^2^ = 0.993 and the limits of agreement, expressed by the accuracy (systematic bias) and the precision (SD), was between +9.45 × 10^−3^ m^−1^ and −6.78 × 10^−3^ m^−1^ for all load levels. The new measuring system offers both good accuracy (1.33 × 10^−3^ m^−1^) and high precision (4.14 × 10^−3^ m^−1^). However, the results are based on quasi-static ski deformations, which means that a transfer into the field is only allowed to a limited extent since the scope of the curvature model has not yet been definitely determined. The high laboratory-related reliability and validity of our novel ski prototype featuring PyzoFlex^®^ technology make it a potential candidate for on-snow application such as smart skiing equipment.

## 1. Introduction

### 1.1. Ski Deflection in Alpine Skiing

With the development of shaped skis in the late 1990s, performing carved turns has become possible. A carved turn is defined by minimal or no lateral ski displacement relative to the track, and therefore, each point along the ski edge follows the path of the proceeding one [1,2,3,4]. In contrast, during a skidding turn, a point along the edge does not follow the path of the proceeding one but slides sideways along the slope [1,2,3,4]. Skidded and carved turns differ not only in terms of ski trajectory but also regarding ski deflection. While in carved turns, ski deflection is a prerequisite to meet the above definition, in skidded turns, the ski does not necessarily have to be deflected. For practical reasons, the skiing motion is often described either as skidding or carving, but carving and skidding are not dichotomous processes; rather, both can occur at the same time in different segments of the ski’s length [3]. To represent the inhomogeneous deflection progression, it would thus be an oversimplification to fit a circle with a constant radius into the deflection line of the ski. One would need several circles with different radius instead. In addition to the segmental differences in deflection along the ski, the temporal sequence of deflection varies in short time periods within a turning phase [5]. However, temporal and segmental changes in deflection during alpine skiing give an essential insight into how the ski-snow interaction proceeds and provides information about the quality of a turn.

### 1.2. Status Quo and Limitations of Research in Measuring Ski Deflection

Standardized laboratory measurements are suitable for a more detailed examination of ski deflection. One of the most often applied methodologies to deform a ski under standardized conditions is the application of a 3-point bending test. To quantify the deflection profile of the ski, the vertical displacement (w) [6,7,8,9,10], the angular deformation [11] or the curvature [12] are typically measured. Most applications measure the vertical deflection by using displacement transducers [6,7], linear potentiometers [8], laser transducers [9] or linear variable differential transformers [10]. Additionally, motion capture systems are used to detect w [10]. Less common is the angular deformation measurement with an optical encoder [11] or the curvature detection with a digital radius gauge, whereby the last was conducted for snowboards [12]. From a mathematical perspective, the angle (gradient) corresponds to the first derivative (dw/dx) and the curvature to the second derivative $\left(d^{2}w/dx^{2}\right)$ of the deflection line. Therefore, these physical quantities can be transformed into each other under consideration of the boundary conditions. A detailed consideration or analysis of the angular or curvature progression has not been carried out so far, rather it was used implicitly to calculate the bending stiffness progression, which is a fundamental design parameter of alpine skis [6,7,9,10,11,12,13]. To operationalize the segmental ski deflection, w(x) is not sufficient because only a qualitative description would be possible. To determine the segment where the largest deflection occurs in a quantitative way, either the angle or the curvature is needed. In summary, there are a number of studies that have dealt with standardized ski deflection in the laboratory, but the exact deflection characteristic along the ski was not analyzed in detail.

The predominant sensors that have been used up to now to determine the ski’s deflection during alpine skiing are strain gauges [5,14,15,16,17,18,19]. In several articles, an aluminum deflection sensor beam with integrated strain cells was applied to the upper surface of the ski [5,15,16,17]. A mathematical method was used to model the deflection of the ski’s running surface based on measured bending and torsion [17]. The snow measurement showed that the deflection consists predominantly of bending and that the torsional deformation has only a minor influence on the shape of the running surface [17]. It was indicated that the ski deflection varied within a few milliseconds [5], and the inner ski was less bent than the outer ski [17]. The construction of the ski with the aluminum deflection sensor beam is very complex, which affects the dynamic properties of the ski [17] and makes a series implementation not possible. Fauve et al. [19] analyzed the influence of snow penetration strength and loading on ski deflection and Schindelwig et al. [18] the influence of bending and torsional stiffness with regard to ski deflection. In another study, the captured turn radii from a differential global navigation satellite system were compared with the calculated radii from strain gauges [14]. A smart ski prototype with a feedback system was also equipped with bending sensors, although the signal was not calibrated and thus only has relative validity [20].

Research about ski deflection, in general, is scarce and, more specifically, the analysis of ski deflection in view of local and temporal variations. Yoneyama et al. [17] differentiated between front and rear ski segments by calculating dw/dx in relation to the boot center coordinate system. However, the gradient was not considered further in the analysis. Other authors either described the segmental deflection of the ski running surface qualitatively with  w(x) [5,16,19] or calculated the overall bending radius ($R=\bar{1/(d^{2}w/dx^{2}}$) along the running surface [14]. The temporal deflection was also considered only at a rudimentary level. So far, only proof of concept studies with superficial validation approaches was provided for the strain gauge-based prototypes, but no concrete applications or commercial spin-offs took place. This may be related to the complex serial implementation of strain gauge sensors. From a scientific point of view, the strain gauge-based systems were not even tested for reliability or validity. In summary, some promising approaches to quantify the ski deflection during alpine skiing were pursued [5,14,15,16,17,18,19,20], but there is a paucity of approaches regarding segmental and temporal deflection with respect to scientific evaluation. Consequently, no useful real-world application is currently available.

### 1.3. Novel Prototype for Ski Deflection Detection

Here, we present a new approach for ski deflection measurement relying on PyzoFlex**^®^** technology (www.pyzoflex.com accessed on 15 July 2021, Joanneum Research Forschungsgesellschaft m.b.H, Franz-Pichler Str. 30, 8160 Weiz, Austria). The novel measurement system will be used to resolve the segmental curvature along the ski and obtain a more differentiated picture of the ski deflection. The PyzoFlex**^®^** foils can be produced by a cost-effective screen-printing process, and therefore, the sensors are commercially highly relevant. Different arrangements, sizes and shapes of the sensors can be specified and easily printed. Furthermore, the foils can be readily applied to skis with the help of double-sided adhesive tape. The ski equipment and the dynamic behavior of the skis are hardly influenced by the foils since the sensors are flexible and practically massless. First, rudimentary on-snow proof of concept tests with a simple sensor layout served plausible raw signals. Therefore, the PyzoFlex**^®^** technology seems to have application potential in the context of alpine skiing.

### 1.4. Goals and Research Questions

The aims of this study were to (a) develop a ski prototype with PyzoFlex^®^ technology-based sensor foils, (b) establish a curvature model based on the PyzoFlex^®^ signal applying standardized bending conditions and (c) test the reliability and validity of this model.

## 2. Materials and Methods

### 2.1. Development of the Ski Demonstrator

The PyzoFlex**^®^** sensors are based on a ferroelectric polymer material and are thus intrinsically piezoelectric and pyroelectric. The sensor foil is constructed in a sandwich structure. The basis of the sensor foil is a transparent polyethylene terephthalate substrate (PET), which serves as a carrier for the printed ferroelectric material. The piezoelectric and pyroelectric sensor material Poly(vinylidene fluoride–trifluoroethylene), denoted as P(VDF-TrFE), is located between the top and bottom electrodes [21]. When a stimulus is applied to the sensor, the surface charge density of the sensor material changes, inducing a change of compensation charges in the electrodes.

To determine the ski deflection, an arrangement of 24 single sensor elements within a foil was first considered and designed in a computer-aided design program. The production of the sensor foils with this design was achieved with an industrially screen-printing process with a carrier substrate (PET) of the sensors with 125 µm and the printed layers (base electrode, active material, cover electrode, silver tracks and protective layer) with a total of about 20 µm. The maximum film size to be produced is limited to A3 format due to the available screen-printing equipment. In the case of the ski design, this resulted in three film elements, which were distributed separately over the ski (Figure 1a). The separation of the three sensor elements—2 foil pieces with 9 sensors, 1 foil piece with 6 sensors—was achieved with the use of a Trotec laser cutter. The electrical connections (printed silver lines) were chosen in such a way that the 24 sensors can be wired via three interfaces. The interface between sensor foil and electronics was realized with foil crimps (Crimpflex^®^
www.dico-electronic.de, accessed on 15 July 2021, DICO Electronic GmbH, Rotenbergstraße 1A, 91126 Schwabach, Germany). On the film side, the crimps were directly connected to the silver tracks of the film, and a housing was plugged over the contact terminals. The mating connector was also connected to a flat ribbon cable by means of crimp contacts and a corresponding housing. The second side of the ribbon cable was connected directly to the readout electronics via terminal plugs. The three foil elements have been laminated onto the ski (Atomic Redster G7; length: 1.82 m; radius: 19.6 m) with a high-performance adhesive tape of the 3M™ VHB™ series (VHB 5952) with a thickness of 1.1 mm (Figure 1b).

A development kit (Joanneum Research, Graz, Austria) was used as a data logger to record the charge response of the PyzoFlex^®^ sensors. It was comprised of a Raspberry Pi 3B+ carrying a self-developed hock-up board containing the analog input stages and a multichannel analog to digital converter (ADC). The Raspberry PI communicates with the ADC (23.7 kSPS/channel) on the hock-up board via a serial peripheral interface. A total of 16 optional sensor signals (out of 24) can be read upon amplification by an operational amplifier circuit. To this end, a charge amplifier circuit (charge-to-voltage converter) with a feedback capacity of $C_{f}=20\;\mathrm{nF}$ was chosen. The value of this capacitance and the resulting measuring range were obtained by test measurements on the bending machine (see experimental setup below). The sampling frequency of the system was 215 Hz/channel. The data were transferred via Ethernet to an external device on which self-developed software was used to display the data and store it in an SQLite database.

### 2.2. Development of the Empirical Curvature Model

To render the printed sensor elements piezoelectric, a poling step is needed [22]. During electrical poling, a high voltage signal is applied to the top and bottom electrodes, which aligns the microscopic dipoles of the ferroelectric layer toward the external electric field. As a result of this poling step, the ferroelectric layer possesses a remnant polarization $P_{r}$ with an out-of-plane orientation, i.e., normal to the electrodes’ surfaces (indexed with (3). The value of $P_{r}$ is obtained directly from the poling hysteresis and serves as a benchmark for the piezo- and pyroelectric response of the sensor layer. A variation in the average out-of-plane strain component $s_{33}$ directly leads to a change in the polarization, which can be measured as a charge response Q. This electromechanical coupling is described by the piezoelectric constant $e_{33}$ as follows [23,24]:(1)$$e_{33}=\frac{-Q}{A}\cdot \frac{1}{s_{33}}=-a\cdot P_{r}$$
where A is the interface area between the electrode and the ferroelectric material, and a is a material constant close to unity.

To derive the bending response of the PyzoFlex**^®^** sensors in the ski application, the local bending at the sensor position needs to be related to the induced strain component $s_{33}$. The scheme in Figure 2a depicts the simplified situation with the sensor clamped on the ski. Bending of the ski with a mean bending radius *R* causes a lateral, in-plane strain $s_{11}$ in the piezoelectric layer of
(2)$$s_{11}=\zeta /R,$$
with *ζ* denoting the distance of the ultrathin piezoelectric layer from the neutral axis. The introduced lateral strain translates into a vertical, out-of-plane strain $s_{33}=-\nu_{13}\cdot s_{11}$ with Poisson’s ratio $\nu_{13}$. By combining Equations (1) and (2), the bending charge response is derived as:(3)$$Q=-A\cdot \nu_{13}\cdot a\cdot P_{r}\cdot \zeta \cdot 1/R.$$

Thus, for ideal bending conditions, the charge response is indirectly proportional to the bending radius and directly proportional to the mean curvature $\bar{w_{i}^{\prime \prime }\left(x\right)}=1/R$ (Equation (3). It should be noted that $\nu_{13}\;$ and a are intrinsic material properties and therefore equal for all sensor elements, whereas A=L⋅b depends on the sensor element’s geometry (length L and width b, cf. Figure 2a). Finally, *P_r_* can be obtained from the poling process and varies only slightly among the sensor elements. The only unknown factor is the location of the neutral axis, as it depends on the inner structure and elastic properties of the rather complex ski construction. Furthermore, the ski’s cross-section and mechanics vary along the ski axis, which results in a position-dependent ζ(x). Consequently, to be able to derive the local ski bending curvature from the measured charge response, a calibration procedure is necessary for every combination of ski design, sensor design and alignment, respectively.

The input charge Q(t) is converted by a charge amplifier circuit (Figure 2b) into a corresponding output voltage $\;u_{a}\left(t\right)$. The amplification is determined by $C_{f}$ in the feedback loop. The output voltage is given as
(4)$$u_{a}\left(t\right)=-\frac{Q\left(t\right)}{C_{f}}=\frac{A\cdot \nu_{13}\cdot a\cdot P_{r}\cdot \zeta }{C_{f}}\cdot \bar{w^{\prime \prime }}\left(t\right).$$

No generalized correlation between the PyzoFlex**^®^** sensor signal and the curvature could be found in preliminary experiments, as was expected due to the unknown parameter  ζ(x). Instead, a position-specific correlation between $\bar{w^{\prime \prime }\left(x_{i}\right)}$ and $u_{a}\left(x_{i},t\right)$ for the sensor element at position $x_{i}$ was found. Therefore, each sensor was calibrated under two extreme load conditions. For each sensor a linear regression model was created to derive the mean curvature from the signal in accordance with the physical model (Equation (4)). As calibration factors, the slope *k* and the intercept *d* of the linear function of the form
(5)$$\bar{w^{\prime \prime }}\left(u_{a}\right)=\frac{-C_{f}}{A\cdot \nu_{13}\cdot a\cdot P_{r}\cdot \zeta }\cdot \left(u_{a}-u_{0}\right)=k\cdot u_{a}+d\;$$
were used, where $u_{0}$ denotes a voltage (i.e., charge) baseline offset.

### 2.3. Reliability and Validity Assesment

#### 2.3.1. Experimental Setup

In order to deform the ski in a standardized way and measure the deflection with high precision, a three-point bending test was carried out on a bending machine (Atomic GmbH, Austria). The basic structure is a welded steel construction with a solid granite block, which ensures that the deflection measurement can be carried out free from ambient oscillations and vibrations (Figure 3a). The ski was mounted in the front (support A) and rear (support B) end with rotatable clamps about the y-axis. The force *F* was applied via a trapezoidal spindle linear actuator (Type LMR 03, Servomech S.p.a, Italy) in the middle of the binding system and was monitored by a force sensor (HI U9C, HBM GmbH, Germany). To measure the vertical displacement of the ski, the bending device was equipped with a laser measuring system (LK-H157, Keyence AG, Japan). The laser manufacturer specifies a measuring range of ±40 mm and a repeatability accuracy of 0.25 μm. The laser unit was moved in x-direction by a servo motor (EMGA-60-P-G5-SAS-70, Festo SE & Co. KG, Germany) with an increment of 20 mm and captured the vertical distance (z-axis) to the ski’s underside at each of these points. Since the ski was supported at the front and rear end and the laser unit cannot reach the support ends, the rear 76 mm and the front 190 mm of the 1820 mm ski were not captured. To obtain w(x) over the entire length between the two supports, three data points in the rear and two data points in the front are extrapolated by fitting a second-order polynomial function through the last four measurement points. Consequently, N = 83 data points (78 directly by the laser measurement system and 5 extrapolated) were captured over the ski length of 1650 mm for each deflection (Figure 3b).

To determine the precision of the laser measurement system for the specific experimental setting, the standard deviation (SD) at each measurement point was calculated over 10 repetitions. The maximum SD in the z-direction (laser) was 99.26 μm and in the x-direction (servo motor) 29.88 μm.

Reliability and validity assessments were performed on the instrumented ski described in 2.1, excluding the three sensors in the front since they are located outside support B, and thus, no laser data are available. The remaining 21 sensors were captured in two rounds since the data acquisition device has only 16 inputs; therefore, the measurements were first performed with the outer sensor rows (14 sensors); subsequently, data of the middle sensor row (7 sensors) were recorded. For the study, the ski was deformed with five different load levels and three repetitions at each level (100 N, 110 N, 160 N, 220 N and 230 N). A load of 230 N seems to be sufficiently large, since the vertical deflection in the middle segment is about 60 mm. In comparison, the vertical deformation in exemplary field tests of skis instrumented with strain gauges was a maximum of 40 mm [15,16,17]. Due to the high precision of the laser and the high time required, laser data was only recorded for the measurements of the outer sensor rows. Considering the symmetrical design of the ski and the pure bending load (no torsion), it is assumed that parallel arranged sensors along the y-axis are deformed to the same amount.

#### 2.3.2. Data Processing–Laser Measurement System

For every load level, the average value of each measurement point was calculated over the three repetitions. Based on the detected laser data, the curvature function of the ski was derived. In the ski shape model of Yoneyama et al. [17], the vertical displacement was captured with an increment length of 100 mm along the ski length, and a 10th order polynomial function was used for the interpolation. Due to our smaller increment length of 20 mm and the high accuracy of the laser measurement system, a polynomial function of a higher degree seems to be appropriate to represent the curvature progression in a highly differentiated way, but a polynomial function with a higher order causes oscillation in the edge regions due to Runge’s phenomenon [25]. This problem can be avoided by using cubic spline curves, which are piecewise polynomials of the third order. The cubic spline minimizes the Holladay functional $\left(I\left(w\right)=\int_{a}^{b}{\left(w^{\prime \prime }\left(x\right)\right)}^{2}\cdot dx\right)$, which causes the second derivative to be best approximated. The spline result can, therefore, be seen as an attempt to interpolate w(x) in such a way that minimal curvatures occur between the data points. A cubic spline interpolation through all 83 data points would fit w(x) very well but generates fluctuations in the second numerical derivative and thus increases the relative error in curvature calculation. Therefore, it is not appropriate to lay splines through all 83 data points. Since the sensors with a length of 100 mm detect an average curvature in the respective segment, it seems reasonable to represent the curvature characteristic of the 1820 mm long ski with the help of 18 splines. Therefore, a cubic spline function with 18 splines was placed through the measuring points to interpolate w(x). By calculating the second spatial numerical derivative of w(x), the curvature function of the ski $w^{\prime \prime }\left(x\right)$ can be obtained:(6)$$w^{\prime \prime }\left(x\right)≝\frac{\partial^{2}w}{\partial x^{2}}$$

In order to perform a numerical computation of the mean curvature $\bar{{w_{i}}^{\prime \prime }}$ of the different sensor segments, $s_{i}$, $w^{\prime \prime }\left(x\right)$ was interpolated again with cubic splines so that $\bar{{w_{i}}^{\prime \prime }}$ could be calculated over 10 points. The average value of the spline curve over the interval ($b_{i},a_{i}$) (Figure 4) defines the mean curvature:(7)$$\bar{{w_{i}}^{\prime \prime }}\;≝\;\bar{\frac{\partial^{2}w_{i}}{\partial x^{2}}}=\frac{1}{b_{i}-a_{i}}\overset{b_{i}}{\underset{x=a_{i}}{\sum }}w^{\prime \prime }\left(x\right)\cdot \Delta x$$

For each of the seven segments, $\bar{{w_{i}}^{\prime \prime }}\;$ was calculated. It is assumed that the curvature is constant along the ski width, which means parallel sensors along the y-axis are deformed to the same amount.

#### 2.3.3. Data Processing—PyzoFlex^®^ Sensor System

All PyzoFlex^®^ data recorded during the measurement was calibrated with the empirical curvature model described in Section 2.2. The calibration coefficients k and d (see Equation (5)) were determined using two separate trials at 100 N and 230 N. A 2nd order Butterworth filter was implemented to remove noise from the raw data. In particular, low-frequency and temperature-related signal fluctuations due to the pyroelectric effect are filtered. The high-pass filter with a cut-off frequency of 0.05 Hz was implemented using MATLAB (R2018B, MathWorks, Natick, MA, USA). As the maximum value of the sensor signal is decisive in data-processing, this value was determined by a custom peak detection algorithm in MATLAB.

### 2.4. Statistical Analysis

To determine the performance of the PyzoFlex**^®^** sensors, the aspects of instrument reliability (within the PyzoFlex**^®^** sensor system) and criterion validity (between systems) were investigated. For the statistical analysis, all 21 sensors from segments 1 to 7 were included.

For the reliability assessment, the data were checked statistically (Shapiro-Wilk test) for normality of distribution. Changes in the mean (CIM) of $\bar{w^{\prime \prime }}$ for three load levels (100 N, 160 N and 230 N) were analyzed by the use of a paired sample t-test (level of significance *p* < 0.05). The intra-class correlation coefficient (ICC 3.1) and the typical error of measurement, expressed as the coefficient of variation (CV), were used to assess relative and absolute test-retest reliability. ICC 3.1 was interpreted as not acceptable (≤0.74), good (0.75–0.89) and excellent (≥0.9) [26]. For CIM, CV and ICC 3.1, successive pairs of trials were considered (repetition 1 with repetition 2, repetition 2 with repetition 3).

For the criterion validity between PyzoFlex**^®^** and the laser system, Pearson’s correlation coefficient was used. All load levels were considered for the statistical evaluation. Magnitudes of correlations were rated as *r* < 0.45, impractical; 0.45–0.70, very poor; 0.70–0.85, poor; 0.85–0.95, good; 0.95–0.995, very good; and >0.995, excellent [27]. Additionally, a Bland–Altman plot was used to describe the agreement between the two quantitative measurement methods [28]. To assess the limits of agreement (LoA) (±0.96 SD), the accuracy, expressed by the systematic bias, and the precision, expressed by the SD, were calculated.

The paired sample t-test statistics were performed with IBM SPSS Statistics V.26.0 (SPSS Inc., Chicago, IL, USA), and the other metrics were calculated with a custom spreadsheet [27].

## 3. Results

### 3.1. Descriptive Report

Figure 5 shows the raw data of the laser measurement system at different load levels (100 N, 110 N, 160 N, 220 N and 230 N). Therefore, the mean ±1.96 SD of each measurement point was calculated over three repetitions. Figure 5a shows the vertical displacement of all 83 measurement points, and Figure 5b a detailed view over 10 measurement points.

The segmental mean curvature (mean ±SD) calculated from the calibrated PyzoFlex**^®^** data of the seven middle row sensors (orange and blue points) and the calculated curvature characteristic determined from the data of the laser measuring instrument is represented in Figure 6. The gray large data points correspond to the result of the numerical derivation of the cubic spline function from Equation (6), and the small gray data points are the interpolated points in between. The highest curvature value is observed at sensor position 1 in the rear ski segment. At a load of 230 N, the maximum curvature is 0.26 m^−1^, which is equivalent to a segmental mean radius of 3.85 m. The vertical displacement in the center of the ski is the highest, as shown in Figure 5, but due to the larger bending stiffness in the binding area, the ski is only slightly curved in this region ($\bar{w^{\prime \prime }}=0.$11 m^−1^ (R= 9.09 m) at 230 N).

### 3.2. Instrument Reliability and Accuracy

Table 1 shows test-retest reliability results for both repetition pairs at three different load levels (100 N, 160 N and 230 N). The CIM is between −1.41% (−3.30 × 10^−3^ m^−1^) and 0.50% (0.59 × 10^−3^ m^−1^) (over all comparisons). Significant differences are detectable in two of the six repetition comparisons. The maximum CV value of 1.45% can be noted when comparing repetition 1 and 2 at load level 230 N. ICC 3.1 was very close to 1 (ICC 3.1 > 0.961, *p* < 0.001) for all cases.

### 3.3. Criterion Validity

In Figure 7a, data from the PyzoFlex^®^ sensor system was correlated to the criterion instrument (laser measurement system) to determine validity. The slope of the function as well as the correlation coefficient R² are very close to 1. Figure 7b shows the Bland–Altman plot. The systematic bias was 1.33 × 10^−3^ m^−1^, and SD was 4.14 × 10^−3^ m^−1^. The dotted horizontal lines are at ±1.96 SD and represent the LoA. The points marked in red outside LoA originate from the sensors at segment 7.

## 4. Discussion

The purpose of this study was to develop a novel ski prototype for the detection of segmental ski curvature and to evaluate its test-retest reliability and criterion validity. The PyzoFlex**^®^**-Ski-Prototype could be considered as an accurate and precise instrument since data showed a good association between the reference instrument (laser measurement system) and the sensor system under standardized laboratory conditions.

It was shown that the ski demonstrator provides highly reliable data. Both the CIM and CV were below 1.45%. Furthermore, the ICC 3.1 was consistently close to 1 (all ICC 3.1 > 0.961) and indicated excellent reliability [26]. Two of the six repetition comparisons were significantly different, with the highest significance recorded at 230 N when comparing repetition 2 to 3 (CIM = −1.41%, *p* < 0.001). An analogous reliability test of the laser raw data (vertical displacement) shows that all six repetition comparisons differed extremely little in absolute terms but statistically significantly from each other. For the largest difference mentioned (230 N, repetition 2 vs. 3), the CIM (95% CI) for the laser data was −0.05% (−0.03–−0.06) (*p* < 0.001) and the CV (95% CI) is −0.06% (−0.05–−0.07). The minimal but significant differences in the sensor data speak to the accuracy as the differences are attributable to the laser data.

The curvature calculated from the sensor signal is in good agreement with the data of the high-precision laser measurement system. The systematic bias was less than 1.33 × 10^−3^ m^−1^ and therefore shows high accuracy. Additionally, the precision of the signal, which is determined by the limits of agreement (+9.45 × 10^−3^ m^−1^ and −6.78 × 10^−3^ m^−1^), indicates a high validity. The results from this study were in line with the coefficient of determination (R² = 0.993). But it is noticeable that the precision is systematically lower for the outer segment S_7_. All points marked in red that are above the LoA line in Figure 7b are related to sensors of segment S_7_. Perhaps these small but systematic deviations are due to the extrapolation of data points in the edge region during the laser measurements. This hypothesis is supported by the fact that also at the rear segment S_1_, comparatively larger differences are recognizable (Figure 6). However, there may be other reasons as well.

In previous studies, little attention was paid to the ski curvature. Most authors have dealt with the sensor-based detection of the deflection line w(x) but not with $w^{\prime \prime }\left(x\right)$. In those studies, the sensors were calibrated in the laboratory, but there were no reliability and validity tests indicated. Consequently, no comparison to previously used systems can be drawn. Only one study reported that the measurement system was validated, and the RMS error of the ski shape estimation was 0.011 m in the lab [14].

The PyzoFlex**^®^**-Ski-Prototype presented here has no sensors in the binding area, which also applies to comparable prototypes [14,18,19,20]. There is one ski described that is also capable to detect the deflection in the area of the binding system but using a very complex construction [5,15,16,17]. This design affects the dynamic behavior of the skis [17] and is therefore not suitable for commercial applications (e.g., smart skiing equipment). Of course, in principle, it would be advantageous to acquire sensor data also from the middle segment of the ski, but due to the high bending stiffness, the curvatures occurring in this area are comparatively smaller (see Figure 6). Furthermore, if a skier goes into a backward or forward-leaning position while carving, the curvature will primarily discriminate in the front and rear ski areas and less in the binding area. The skier has no possibility to address only the deflection of the middle ski segment. Therefore, if the deflection of the front and rear ski segments is known, the curvature of the middle segment can be approximated. The integration of the sensor foils into the ski structure over the entire length of the ski might be a future perspective to fully cover the deflection behavior of an entire ski. In the current prototype, it was only possible to attach the sensors to the ski surface. This hypothesis still needs to be checked.

Until now, there are no commercial applications that measure ski deflection during skiing. The ski prototype based on PyzoFlex**^®^** technology opens a potential field of application. Not the least because of the low-cost sensor production by screen printing and the flexible sensor design. This sensor layout can easily be modified with regard to size, shape and arrangement of the sensors. Using a hot press process, it is conceivable that the sensor implementation could be embedded in the manufacturing process of the ski. This process protects the sensors from external influences, and sensors can also be integrated under the binding area. Another advantage over strain gauge-based systems is that the film not only functions as a sensor, but also as a mechanical-electrical energy converter. The film not only functions as a sensor, but also as a mechanical-electrical energy converter. Combined with suitable harvester electronics, an area-related power in the range of several µW·cm^−2^ can be generated [22]. There is already a prototype with sensors mounted on the inside of a car tire. At 30–50 km·h^−1^, enough energy is generated to send the measured tire pressure protocol to the board computer every 30 s [22].

It cannot be answered on the basis of these investigations if the used PyzoFlex sensors are generally more suitable for the curvature measurement compared to other sensor technologies because each sensor system has its advantages and disadvantages. However, it should be mentioned that not one of the various prototypes or sensor systems has succeeded in reaching the market, and there are reasons for this. However, the main difference from comparable studies is that the method of deformation analysis is a completely new approach. The subjective judgement leads to the belief that a bent ski is most curved in the middle. This is a fallacy, as the vertical deflection has little to do with the localization of the highest curvature. The curvature progression of the ski and its change over time provides insights into the quality of a turn. If only the bending line is analyzed, this information cannot be extracted, and this distinguishes this work from the others. To determine segmental curvature maxima (or local radius minima), the second derivative of the bending line must be investigated.

However, the current study has some limitations. All measurements were performed in a quasi-static setting. Therefore, a transfer into the field (i.e., on-snow skiing) is only possible to a certain extent. In skiing, there are multifactorial parameters that influence ski deflection. For example, the influence of ski vibrations different excitation frequencies and torsional deformations on the sensor signal cannot be stated yet based on this study. The sensor layout was chosen in such a way that differences of parallel sensors can be used to detect the segmental torsion angles along the ski. This requires an extension of the bending machine to be able to perform combined loads (torsion and bending). Furthermore, the influence of changes in temperature is not yet clearly defined. In principle, dynamic temperature changes can substantially influence the sensor signal due to the pyroelectric effect. So far, the temperature was assumed to be quasi-static because it changes sufficiently slowly (e.g., the heat generated by ski deformations); therefore, temperature changes are predominantly in the low-frequency range, which means that they can be filtered out well with a high-pass filter. Absolute temperature differences do influence the piezoelectric coupling coefficient, but only very weakly [29]. This dependence must be analyzed in more detail in order to introduce a temperature-dependent calibration factor if necessary. Further measurements and analyses are required to answer these open questions.

## 5. Conclusions

The deflection of the ski is the result of the ski-snow interaction and provides information about the quality of a turn. Until now, there are no commercial applications that measure ski deflection during skiing.

A new ski prototype has been developed to determine the segmental curvature of Alpine skis. The PyzoFlex**^®^** technology-based sensor foils are a valid and reliable measuring instrument to measure the segmental curvature characteristic of alpine skis in a standardized setting.

Validating the curvature model in the field is complex since we do not know of any gold standard measuring instrument on snow. A dynamic validation in the laboratory would be a possibility (e.g., by a programmable bending robot) to apply deformations similar to skiing (cyclic deformation, edging). In perspective, not only cyclic loads but also combined loads (torsion and bending) could be applied to determine the scope of the curvature model and to extend it if necessary. Another approach would be to perform a technical validation in the field. However, further improvements of the prototype are planned for the future (e.g., waterproofing, wireless transmission of sensor data, integration of the sensor foils into the ski structure to protect them).

## Figures and Tables

**Figure 1 sensors-21-04848-f001:**
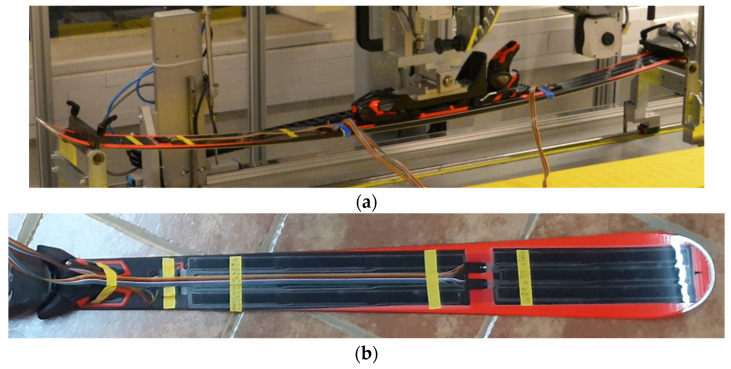
(**a**) Ski prototype with 24 single connected sensors. One foil element was implemented at the rear (nine sensors) and two foil elements at the front (fifteen sensors). (**b**) Detailed view of the sensors in the front ski segment. The sensors were laminated to the ski with a black high-performance adhesive tape (very temperature and UV stable).

**Figure 2 sensors-21-04848-f002:**
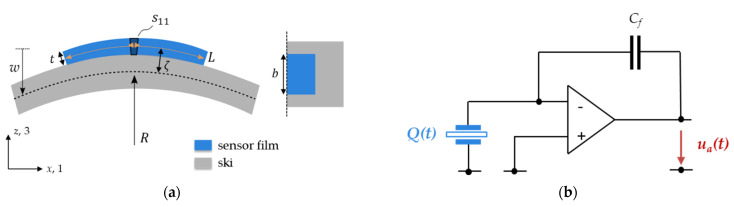
(**a**) The bending scheme of a PyzoFlex**^®^** sensor film mounted on top of the ski at the bending radius $R=1/w^{\prime \prime }\left(x\right)$ (see main text) in cross-section view (left) and top view (right). The bending of the sensor causes a lateral, in-plane strain $s_{11}$ in the piezoelectric layer. The sensor element has length L, width b and thickness t, and the sensitive, piezoelectric layer is located at a radial distance ζ off the neutral axis. (**b**) The sensor generates a charge Q(t), which is converted into a proportional output voltage $u_{a}\left(t\right).$ $C_{f}$ corresponds to the capacitance in the feedback loop.

**Figure 3 sensors-21-04848-f003:**
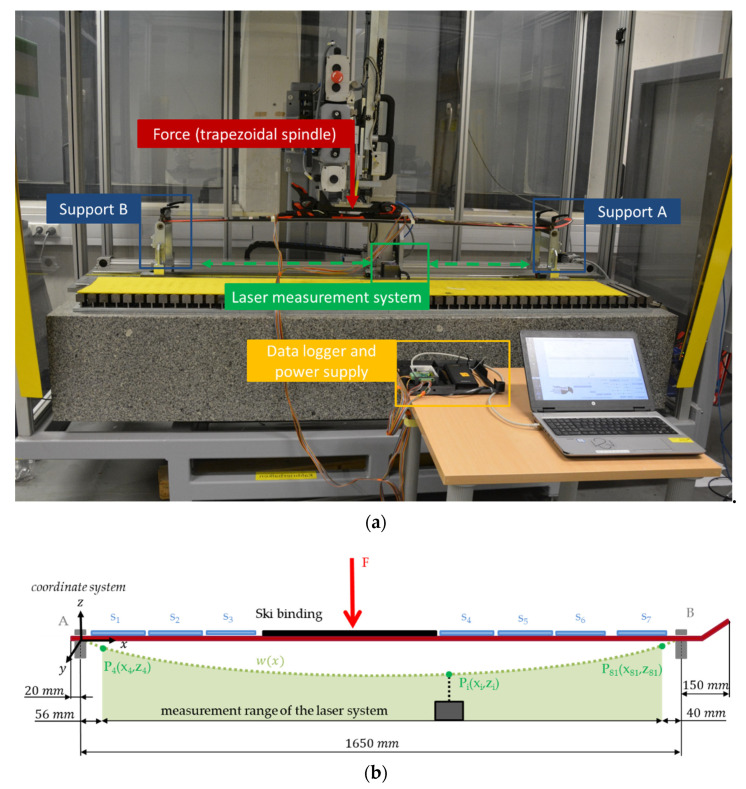
Three-point bending test with an integrated laser measurement system (Atomic GmbH); (**a**) picture with the corresponding experimental components; (**b**) schematic drawing of the experimental setup. N = 83 data points were captured over a length of 1650 mm (78 points directly by the laser measurement system ($P_{4}\left(x_{4},z_{4}\right)$ to $P_{81}\left(z_{81},z_{81}\right)$ ) and 5 points extrapolated).

**Figure 4 sensors-21-04848-f004:**
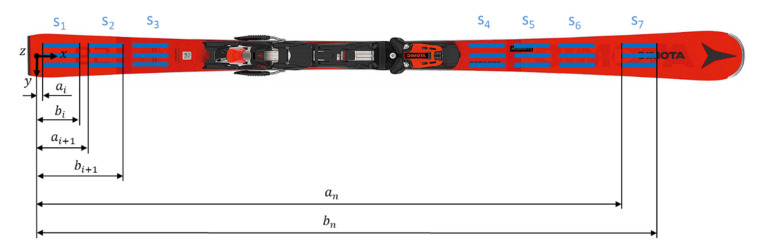
The ski instrumented with PyzoFlex**^®^** technology-based sensors.

**Figure 5 sensors-21-04848-f005:**
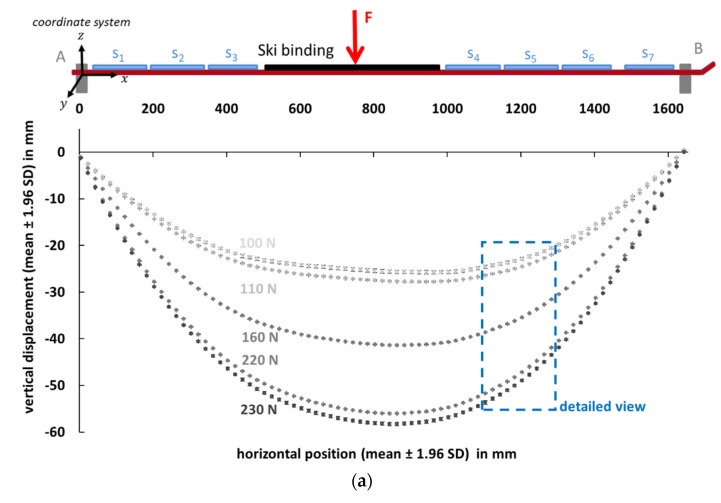
Raw data from the laser measurement system. Vertical displacement (mean ±1.96 SD) vs. horizontal position (mean ±1.96 SD) over three repetitions at different load levels (100 N, 110 N, 160 N, 220 N and 230 N). (**a**) All 83 measuring points; (**b**) Detailed view over 10 measuring points.

**Figure 6 sensors-21-04848-f006:**
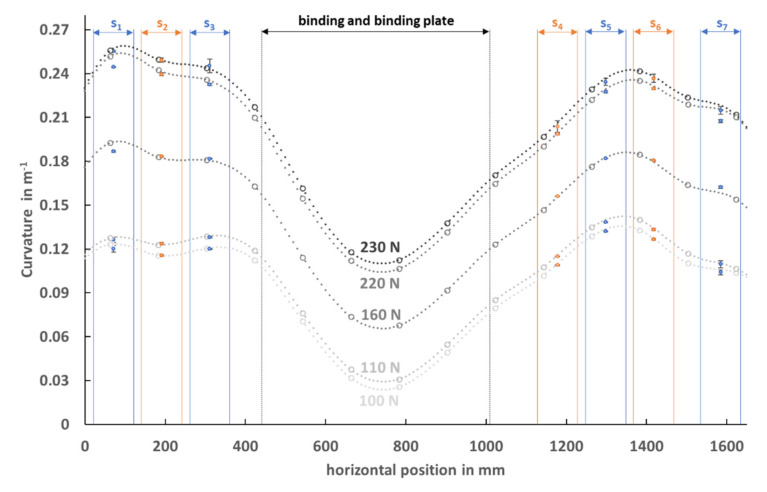
The orange and blue data points represent the calculated mean curvature from the PyzoFlex^®^ data (mean ± SD) of the middle sensor row for the corresponding segments (S_1_ to S_7_). The large gray data points represent the curvature progression calculated from the laser data at different load levels (100 N, 110 N, 160 N, 220 N and 230 N). The interpolated points (small gray data dots) were used for the numerical calculation of the mean segmental curvature. Note: There are no sensors in the binding area, and for the sake of completeness, the data from the laser measurement system are displayed.

**Figure 7 sensors-21-04848-f007:**
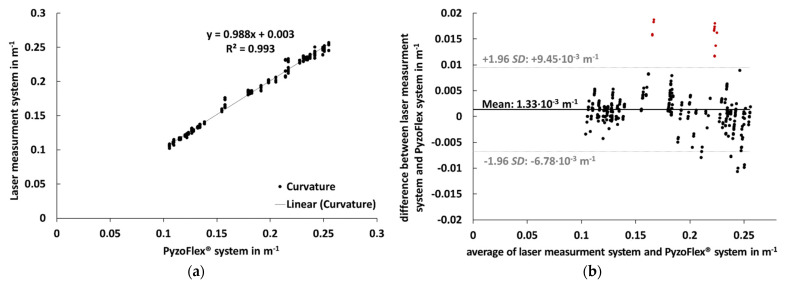
Left (**a**): The correlation between curvature (m^−1^) measured by laser measurement system (criterion instrument) and PyzoFlex^®^ sensor system. Right (**b**): Bland–Altman plot showing the difference against the average of the laser measurement system and PyzoFlex^®^ sensor system with limits of agreement (LoA) (dotted lines). SD: Standard deviation.

**Table 1 sensors-21-04848-t001:** Analysis of instrument reliability from the sensor system.

		*p* (*t*-Test)	CIM (%, 95% CI)	CV (%, 95% CI)	ICC 3.1 (95% CI)
100 N	Repetition 1 vs. 2	n.s.	0.50 (−0.08–1.07)	1.08 (0.86–1.47)	0.982 (0.962–0.991)
	Repetition 2 vs. 3	n.s.	0.39 (0.05–0.73)	0.63 (0.51–0.86)	0.994 (0.987–0.997)
160 N	Repetition 1 vs. 2	<0.01	−0.46 (−0.70–0.23)	0.44 (0.35–0.60)	0.995 (0.990–0.998)
	Repetition 2 vs. 3	n.s.	−0.26 (−0.80–0.27)	1.01 (0.81–1.37)	0.977 (0.952–0.989)
230 N	Repetition 1 vs. 2	n.s.	0.32 (−0.45–1.09)	1.45 (1.16–1.97)	0.961 (0.920–0.982)
	Repetition 2 vs. 3	<0.001	−1.41 (−1.86–0.95)	0.86 (0.69–1.17)	0.983 (0.966–0.992)

CIM: change in mean; CI: confidence interval; CV: coefficient of variance; ICC 3.1: intraclass correlation coefficient.

## Data Availability

The data presented in this study are available on request from the corresponding author.

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
