# Peer review of "A Novel Sensor Foil to Measure Ski Deflections: Development and Validation of a Curvature Model"

_sensors, 2021, doi:10.3390/s21144848_

Round 1

Reviewer 1 Report

I like the article very much , because it is a clear trial to define a new Method to measure ski deflections.

The introduchtion is to long, even for readers, who would first see, what the method is good for. So, I would create a new capitel 2 called "Staus quo of research" to show all the different research areas from literature.

The article is a clear description of a new approach for reserach in that area with all the necessary mathematical formulars (very good!)

The limitations of the study (a quasi-static setting) should be mentioned in the abstract.

Reviewer 2 Report

This article has well described a unique approach for detecting ski deflections by the printable foil sensor of PyzoFlex®. Also the author has clearly described the limitation of the present study in Discussion. e.g. the mechanical interference of the torsion, and the influence of the temperature change because the pyroelectric is sensitive to the temperature. It seems very promising research with further investigations. In order to enhance the value of this article, please consider following review comments.

  1.  Introduction seems to be lengthy by describing the detailed history of the ski deflection measurement. It should be shortened by describing only important things.
  2. P(VDF-TrFE) (poly(vinylidene fluoride–trifluoroethylene)) is strange and complex notation. It should be to change to Poly(vinylidene fluoride–trifluoroethylene) denoted as P(VDF-TrFE).
  3. What does the maximum load of 230N represent? It seems to be small as considering the skier’s body weight and dynamic alpine skiing.
  4. Line 312: ‘p<.05’ should be replaced to ‘p<0.05’ to unify notation and to avoid misreading.
  5. ‘Conclusions’ is very short. Some important results should be included.

Reviewer 3 Report

In this paper,   a new approach for ski deflection measurement relying on  PyzoFlex® technology is presented. The  measurement system will be used to resolve the segmental curvature along the ski and obtain a more differentiated picture of the ski deflection. Some First rudimentary on snow proof of  concept tests with a simple sensor layout served plausible raw signals. The aims of this study were to a) develop a ski prototype with PyzoFlex® technology based sensor foils, b) establish a curvature model based on the PyzoFlex® signal applying  standardized bending conditions and c) test the reliability and validity of this model

The paper is well written and straightforward.  The following comments must be improved and clarified.

-The reviewer does not see clearly how the authors guarantee the measurement of curvature between 400 and 1000 without having the corresponding PyzoFlex® sensors there.

-The conclusions are very brief, The authors must expand the conclusions.  They must also clearly include the advantages of the proposed solution compared to those existing in the market

-How the authors intend to implement this system in real alpine skiing conditions and how real conditions can affect the measurement system

-What are the advantages of the proposed solution compared to others that are already available to measure ski's deflection during alpine skiing and  smart ski prototype with a feedback system

Round 2

Reviewer 3 Report

 Accept in present form